# Understanding E-Scooter Incidents Patterns in Street Network Perspective: A Case Study of Travis County, Texas

Junfeng Jiao [ID], Shunhua Bai [ID] and Seung Jun Choi *[ID]

Urban Information Lab., The School of Architecture, The University of Texas at Austin, Austin, TX 78705, USA; jjiao@austin.utexas.edu (J.J.); shunhua@utexas.edu (S.B.)
* Correspondence: jun.choi@utexas.edu

**Abstract:** Dockless electric scooter (E-scooters) services have emerged in the United States as an alternative form of micro transit in the past few years. With the increasing popularity of E-scooters, it is important for cities to manage their usage to create and maintain safe urban environments. However, E-scooter safety in U.S. urban environments remains unexplored due to the lack of traffic and crash data related to E-scooters. Our study objective is to better understand E-scooter crashes from a street network perspective. New parcel level street network data are obtained from Zillow and curated in Geographic Information System (GIS). We conducted local Moran's I and independent Z-test to compare where and how the street network that involves E-scooter crash differs spatially with traffic incidents. The analysis results show that there is a spatial correlation between E-scooter crashes and traffic incidents. Nevertheless, E-scooter crashes do not fully replicate characteristics of traffic incidents. Compared to traffic incidents, E-scooter incidents tend to occur adjacent to traffic signals and on primary roads.

**Keywords:** e-scooter safety; micro-mobility; street network

## 1. Introduction

Shared micro-mobility has rapidly increased in popularity as an alternative transportation mode in the United States [1–3]. Shared micro-mobility is a user-oriented transportation mode that is easily accessible [4]. Types of shared micro-mobility services include station-based bike-sharing, dockless bike-sharing, and E-scooter sharing. E-scooter services have emerged as a commonly used micro-mobility mode in cities [2,5–7]. The first E-scooter sharing program was launched in the U.S. in September 2017 [8]. E-scooter services have become one of the most popular non-automotive alternatives for people to travel short distances in urban downtowns and universities [3,9].

COVID-19 has brought new safety concerns and challenges that impact traditional travel safety measures and practices. For instance, drastic increases of reported COVID-19 cases caused people to avoid public transportation to prevent viral transmission [10]. The unprecedented threat of COVID-19 stimulated favorable opinion of micro-mobility usage for traveling during the pandemic [11]. Similarly, Teixeira and Lopes found public transportation users utilized micro-mobility options to reduce their health risks [12]. Micro-mobility allows users to access their desired destinations while maintaining social distance from other users and people. As more users are willing to use micro-mobility, micro-mobility will gain greater public attention as an alternative transportation mode post-pandemic. It is imperative for transportation planners and researchers to continuously and actively scrutinize different aspects of shared micro-mobility programs such as travel behaviors and public safety.

In contrast to the amount of shared micro-mobility research on travel behaviors, there is scarce literature on E-scooter safety due to the lack of data. Although researchers have used survey-based methods [13] and data mining techniques [2] to identify scooter-related

injuries and crashes, it is difficult for planners to contextualize the safety issues without sufficient empirical evidence.

Past research regarding E-scooters and dockless mobility has identified the usage and travel patterns of E-scooter users [5,9,14] and characteristics of E-scooter user injuries [6,15]. Shah et al. proved that E-scooter crash characteristics do not fully replicate the features of other shared micro-mobility crashes [7]. E-scooter users experience more severe vibration events than bike riding in the event of a crash [6]. However, these studies did not incorporate street networks in their analysis and instead focused on E-scooter ridership rather than E-scooter associated injuries. Street networks are the imperative urban component associated with transportation crashes and road safety in cities [16,17]. Understanding E-scooter associated injuries or crashes from a street network perspective remains a vital task in order to accurately apprehend the characteristics of E-scooter crashes.

## 2. Literature Review

Dockless electric scooter (E-scooters) services have emerged in the United States as an alternative form of micro transit in the past few years as they allow users to travel short distances quickly [3]. Due to the increasing prominence of micro-mobility in the United States, E-scooters have emerged as a new micro-mobility mode to fulfill increasing short distance and last mile travel demand [6]. E-scooter services are a single user-oriented micro-mobility option that can support and benefit other transportation modes, such as rail and bus [5]. Other benefits of riding E-scooters include reduced greenhouse gas emissions and automobile congestion [1]. Nevertheless, there is still no universal consensus on how to manage E-scooter services in urban environments.

E-scooter services are primarily utilized for recreational purposes [13]. In Austin, TX, our spatial analysis showed that E-scooter usage is concentrated in the downtown and university campus areas [9]. In addition, greater accessibility, land-use composition, and proximity to the city center were positively correlated with higher E-scooter ridership [9].

E-scooter associated injuries are believed to be more serious for users and increase the burden of emergency rooms and other ER related health services [13]. Another problem with E-scooter-associated injuries is that E-scooter crashes cause injuries to E-scooter riders and threaten adjacent street users. Due to the compact size of an E-scooter, E-scooter riders can freely utilize pedestrian streets, and pavements, which result in tensions with and risks to pedestrians [5]. When E-scooters were first introduced, the media did not give full attention to E-scooter-associated injuries. However, reports of E-scooter fatalities and crashes have increased [1].

In 2018, Austin Public Health conducted research to identify the characteristics of E-scooter-related injuries. Among 190 surveyed users, nearly half (48%) had injuries. 70% of injured users had upper limb injuries, such as the shoulder, arm, wrist, and hands, and 55% of injured users had injuries in the lower limbs. One-third of injured riders experienced bone fractures. The results are plausible because the rate of protective gear usage is low among E-scooter riders [13].

E-scooter crashes do not equal other transportation mode crashes [7]. In addition, higher travel speed is likely associated with E-scooter injuries of greater severity [15]. Injuries commonly occurred on the sidewalk or roads, and riders injured on roadways are twice as likely to sustain severe injury than those injured elsewhere [15]. In addition, E-scooter injuries in Austin suggest that E-scooter related injuries tend to occur more often on the weekend than on weekdays and during working hours, between 9 a.m. and 11 a.m. and from 1 p.m. to 4 p.m. [18].

While there are extensive datasets on motor vehicle crashes in the U.S., there are no public E-scooter crash datasets as of this study, making it difficult to run in-depth analytics [2,6,13]. Therefore, the present study's findings do not provide decisive findings concerning E-scooter crashes or injuries. However, it suggests a new research framework to better understand E-scooter injuries and crashes from a street network perspective.

The present study contributes to filling the research gap of past studies by identifying the characteristics of E-scooter incidents in a street network perspective, confirming previous E-scooter travel patterns, and understanding the difference between E-scooter incidents compared to motor vehicle crash incidents.

## 3. Materials and Methods

### 3.1. Study Area

The study area is Travis County, Austin, TX, USA. Zillow neighborhood is the geographical unit used for the analysis. There are 61 Zillow neighborhoods. Zillow is a well-known real estate market platform in the U.S. that offers sales and listings of properties with thorough comparisons. Contrary to census boundaries such as Census Tracts, Zillow publicized their neighborhood boundaries experienced by local housing markets, enabling a new local perspective analysis. Therefore, Zillow neighborhood boundaries do not fully agree with Census-based geographies.

By using Zillow-defined neighborhoods, we can distinguish areas with different characters by their neighborhood names. For instance, North University and UT-Austin in Figure 1 are two separated neighborhoods in one Census Tract, where the former is mainly for student housing. The latter is the main campus. Generally, it goes within the micro-mobility service boundary in the City of Austin (COA). Based on the record from COA during the study period [19], compared to population density, there is relatively greater E-scooter ridership in downtown, park, and university neighborhoods than other service areas.

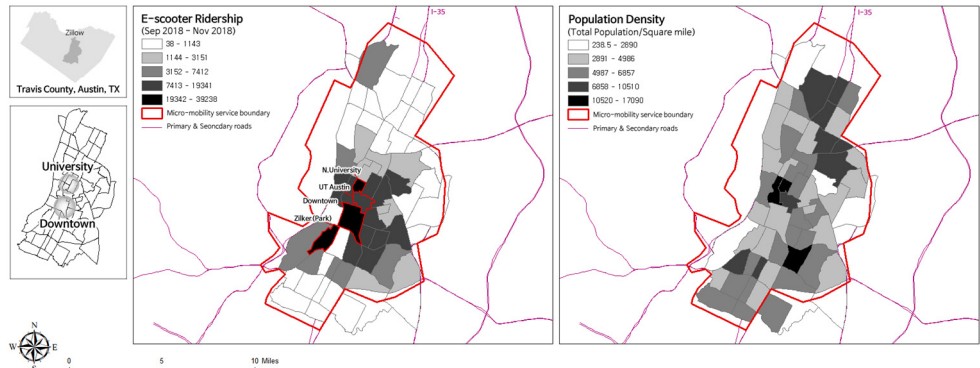

**Figure 1.** Study area.

### 3.2. Data

The present study analyzed Austin, Texas, USA. The study period was from September to November 2018, which consists of a total of 90 days. The research objective was to understand the E-scooter incident from a street network perspective and its occurrence.

E-scooter crash data were obtained from Patch, an American local news platform with a spatial location in Austin, TX, from September to November 2018 [20]. Patch obtained data from Austin–Travis County EMS data. They tracked a total of 166 E-scooter crashes and confirmed that 60 percent of scooter accidents occurred near downtown areas during the study period [20]. The dataset offers location information of an accident, including longitude and latitude, and other information such as contact data, gender of the E-scooter user, injury description, and hospital information. We web scrawled the E-scooter crash map publicized by Patch each by each. The scrawled E-scooter crash with individual longitude and latitude were imported in GIS. We assumed the E-scooter crash outside the E-scooter service boundary as outliers and removed them. As a result, from 166 E-scooter crash records from Patch, a total of 156 E-scooter crash records were used.

Besides E-scooter accidents, we collected land use and built environment data, traffic incidents, and E-scooter ridership data during the same period from the COA website to examine if E-scooter accidents were observed where traffic accidents frequently happened.

Land use and built environment are based on Land Use Inventory Map publicized by COA. It contains land use ratio of single-family, mobile homes, duplexes, apartments, commercial, mixed-use, office, manufacturing, warehousing, miscellaneous industrial, landfills, parks, cultural, transportation facilities, water, agricultural, undeveloped, and other zoning codes by each parcel. Using spatial join and overlay tools in GIS, we categorized the land use data into seven and summed up for each category per geographic unit of the study: residential, commercial, office, industrial, public, open space, and transportation related land use. Residential land use includes single-family, mobile homes, sizeable lot-single family, three/fourplexes, and apartment/condo. Commercial land use includes commercial and mixed-use. Office land use only includes office. Industrial land use includes manufacturing, warehousing, and miscellaneous industrial land use. Public land use includes landfills, semi-institutional housing (housing for mentally and physically ill), government services, education, and cemetery. Open space land use includes parks/greenbelts and common acres. Lastly, transportation related land use includes railroad, aviation, transportation facilities, and streets and roads.

Traffic incidents data offer address information with longitude and latitude. During the study period and within the study boundary, 7715 traffic incidents were reported. E-scooter ridership data is provided for each census tract that falls into the E-scooter service boundary in the City of Austin. It contains trip duration, distance, start time, and end time with a designated E-scooter ID. Due to the large set of datasets, we used Python Pandas to import the data and grouped them daily per tract and hourly per tract. During the study period, 1,067,298 E-scooter riderships were reported with an average trip duration of 12 min (11.60) for each E-scooter. An average of 191 traffic incidents and an average of 11,728 E-scooter riderships were reported daily. For spatial analysis, the tract was areal interpolated into the Zillow neighborhood boundary in GIS.

Zillow, a real estate platform, provides geographic neighborhood boundary and street networks with nodes and edges. Zillow street network data include nodes and edges. There are 11,149 Street Nodes (SN) and 15,179 Street Edges (SE) within the study area. SN offers motorway junction, traffic signal, turning circle, and loop information. SE includes the type of street network, number of lanes, max-speed, name, length, and one-way information. However, several data had most of their information as N/A. For instance, only 1025 (7%) SE had max-speed information. Similarly, only 1579 SE (10%) includes several lanes. Therefore, this information was hardly included in the analysis.

Figure 2 demonstrates the open-source data used in the present study. Kernel density results show that E-scooter incidents show similar greater density in downtown and near university neighborhoods as traffic incidents. Though kernel density of traffic incidents tends to follow the main road linked to downtown and university, it was most dense in the downtown neighborhood.

Now with these data, we were able to fill the gaps of past research. Specifically, the present study explored E-scooter rider injuries from a street network perspective and identified characteristics of the street network associated with E-scooter injuries compared to motor crash ones. Moreover, we verified the past studies' findings regarding E-scooter ridership and its travel patterns.

Lastly, a two-sample independent Z-test was conducted to determine the statistical difference between E-scooter incidents involving street networks and motor crashes involving ones.

Table 1 summarizes variables used for independent Z-test and descriptive statistics of each variable. Land use and built environment, residential, commercial, office, industrial, public, open space, and transportation-related land use were collected. For street edges, length of the street, dummy of one-way, link, living street, motorway, primary, residential, secondary, territory, trunk, and unclassified road type were used. The type of street edge was provided in one single column as a nominal factor. Therefore, this information was assigned to individual dummies as a numeric variable. Regarding street nodes, a total number of nodes, such as motorway junction, traffic signal, turning circle, and stop sign,

was used. Street nodes are a single dot that contains nominal factor which explains what the node refers to, such as motorway junction, traffic signal, tuning circle, and stop sign. Each node was reclassified by its types and encoded as numeric value one so that we could add up the number of motorway junctions, traffic signals, tuning circles, and stop signs adjacent to crash data points. Therefore, the street nodes variables used in Z-test refer to the total number of nodes for each type inside the designated distance from the individual crash points.

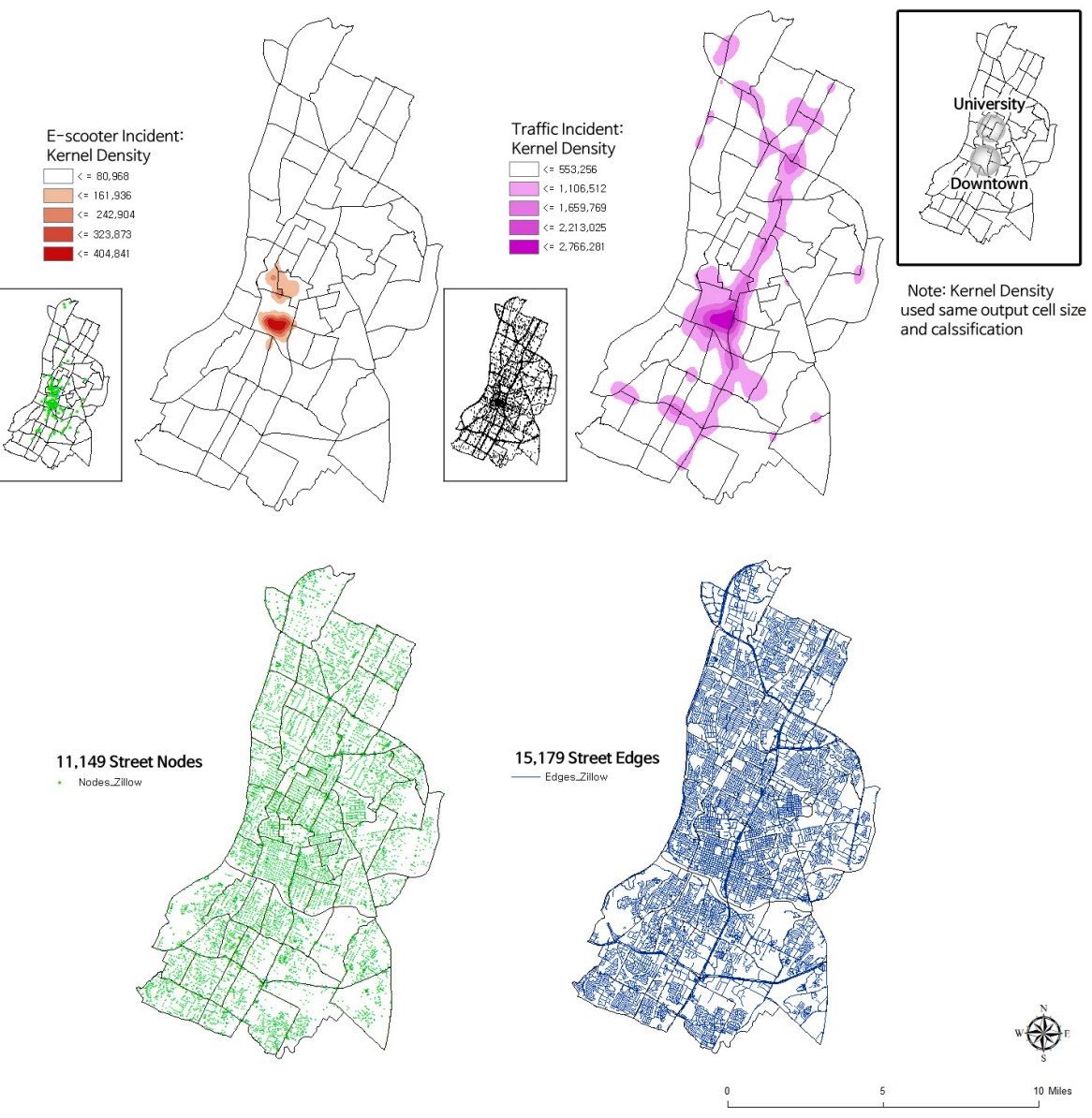

**Figure 2.** Collected open-source data.

**Table 1.** Variables and descriptive statistics for Z-test.

| Variable | | Unit | Count | Mean | Std | Min | Max |
|---|---|---|---|---|---|---|---|
| Land Use and Built Environment (COA, 2012) | Residential | ratio | 1566 | 0.432 | 0.286 | 0.000 | 0.916 |
| | Commercial | ratio | 1566 | 0.080 | 0.126 | 0.000 | 0.668 |
| | Office | ratio | 1566 | 0.041 | 0.094 | 0.000 | 1.000 |
| | Industrial | ratio | 1566 | 0.033 | 0.101 | 0.000 | 0.807 |
| | Public | ratio | 1566 | 0.051 | 0.115 | 0.000 | 0.863 |
| | OpenSpace | ratio | 1566 | 0.033 | 0.087 | 0.000 | 0.827 |
| | Transportation Related | ratio | 1566 | 0.011 | 0.040 | 0.000 | 0.443 |
| Street Edges (Zillow, 2017) | Length | meter | 1566 | 168.636 | 131.382 | 3.560 | 1015.575 |
| | One-way | dummy * | 1566 | 0.159 | 0.366 | 0.000 | 1.000 |
| | Link | dummy * | 1566 | 0.028 | 0.165 | 0.000 | 1.000 |
| | Living Street | dummy * | 1566 | 0.001 | 0.025 | 0.000 | 1.000 |
| | Motorway | dummy * | 1566 | 0.011 | 0.104 | 0.000 | 1.000 |
| | Primary | dummy * | 1566 | 0.050 | 0.218 | 0.000 | 1.000 |
| | Residential | dummy * | 1566 | 0.538 | 0.499 | 0.000 | 1.000 |
| | Secondary | dummy * | 1566 | 0.205 | 0.404 | 0.000 | 1.000 |
| | Territory | dummy * | 1566 | 0.000 | 0.000 | 0.000 | 0.000 |
| | Trunk | dummy * | 1566 | 0.003 | 0.056 | 0.000 | 1.000 |
| | Unclassified | dummy * | 1566 | 0.010 | 0.101 | 0.000 | 1.000 |
| Street Nodes (Zillow, 2017) | Total Nodes | count | 1566 | 2.334 | 2.495 | 0.000 | 27.000 |
| | Motorway Junction | count | 1566 | 0.033 | 0.179 | 0.000 | 1.000 |
| | Traffic Signal | count | 1566 | 0.131 | 0.337 | 0.000 | 1.000 |
| | Turning Circle | count | 1566 | 0.050 | 0.219 | 0.000 | 1.000 |
| | Stop Sign | count | 1566 | 0.003 | 0.050 | 0.000 | 1.000 |

* Dummy compose 0 and 1.

### 3.3. Methods

The study phase goes through three stages. First, time series analysis has been performed to reveal the travel pattern of E-scooter ridership, E-scooter incidents, and traffic incidents throughout time. Here we also visualized correlation matrix using Python libraries, including Matplotlib and Seaborn. Matplotlib and Seaborn are effective libraries for visualizing data. Second, spatial autocorrelation (local Moran's $I$) was used to identify the spatial pattern of abovementioned three major variables.

Local Moran's $I$ is a tool that determines spatial cluster and outlier with statistical significance, and it is only reliable when the input feature contains at least 30 features [21]. It is a useful tool to identify hotspots and relevant test statistics used for spatial data analysis [21–23]. It calculates local Moran's $I$ value, a z-score, p-value, and a numeric code refer to a type of cluster with statistical significance. The local Moran's $I$ follows the equation below. The high positive local Moran's $I$ value implies the location has similar values as its neighbors. On the other hand, high negative local Moran's $I$ value means the location is containing distinct values and forms spatial outliers. In transportation accident

research, local Moran's *I* has been used to detect accident hot spots [24–27]. Table 2 discerns a variable used in the analysis.

$$I_i = \frac{z_i - \bar{z}}{\sigma^2} \sum_{j=1, j \neq i}^{n} \left[ w_{ij(z_j - \bar{z})} \right]$$

where $z_i$: value of variable $z$ at location $i$,

$\bar{z}$: average value of $z$ with the sample number of $n$,
$z_j$: the value of the variable $z$ with all the other location (where $j \neq i$),
$\sigma^2$: the variance of variable $z$,
$w_{ij}$: a weight which can be defined as the inverse of the distance between location $i$ and $j$.

Weight can also be determined using a distance band: samples within a distance band are given the same weight, while those outside the distance band are given the weight of 0 [22].

**Table 2.** Variables used for time series analysis and spatial statistics.

| Variable | | Unit | Source | Year |
|---|---|---|---|---|
| Crash | E-scooter | Count | Patch | 2018 |
| | Traffic Incident | Count | COA | 2018 |
| Ridership | E-scooter | Count | COA | 2018 |
| Study Period: September–November 2018 | | | | |

Third, a two-sample independent Z-test was conducted to determine the statistical difference between E-scooter incidents involving street networks and motor crashes involving ones. Street network refers to surrounding street edges, nodes, land use, and built environment of each incident. Z-test statically tests means of samples and determines significant differences between groups. It is best used when there are more than 30 samples, assuming that when there is a more significant number of samples the samples are believed to be generally distributed under the central limit theorem. The Z-test equation is described below.

$$Z = \frac{\underline{X_E} - \underline{X_T}}{\sqrt{\frac{\left( SE_{X_E} \right)^2}{N_E} + \frac{\left( SE_{X_T} \right)^2}{N_T}}} \tag{1}$$

where $\underline{X_E}$: average of E-scooter injury associated street network,

$\underline{X_T}$: average of traffic incident associated street network,
$\overline{SE_{X_E}}$: standard error E-scooter injury associated street network,
$SE_{X_T}$: standard error of traffic incident associated street network,
$N_E$: number of E-scooter injury associated street network,
$N_T$: number of traffic incident associated street network.

The unit of analysis for independent Z-test is a street network that only involves either E-scooter incidents or motor crashes. To add incident information into the individual street network, we had to use ArcGIS. Data curation goes through three steps: First, individual crash records were geocoded using ArcMap 10.8. Second, crash records were spatially merged to the overlapping street network. Third, around a 200 m buffer centered at each traffic incident location was created to capture the number of adjacent street nodes' information, built environment, and land use features and measure street edges. A round buffer enables counting and measuring the density of built environment variables [28]. Two hundred meters is a two-to-three-minute walking distance and best suited for preventing the creation of outliers for our analysis.

In total, 1566 street networks were used. A total of 149 street networks (10%) composed E-scooter incidents and 1417 street networks (90%) were associated with traffic incidents. For technological support, Stata 16, ArcMap 10.8, and Tableau Desktop were used.

## 4. Results

### 4.1. Time Series Analysis

The descriptive analysis of hourly patterns accumulated for the study period is shown in Figure 3. It is the result of hourly grouped and summed up E-scooter ridership, incidents, and traffic incidents using Python Pandas. E-scooter ridership gradually increases from 6 a.m., hits its peak during the afternoon (12 p.m. to 5 p.m.), and incrementally decreases at nighttime (after 5 p.m.). Traffic incidents gradually accumulated starting from 9 a.m. and hit their peak at 9 p.m. However, contrary to E-scooter ridership, greater cases of traffic incidents were reported even in evening hours. E-scooter incidents tend to take place more during the afternoon or night hours than morning or evening hours. The lowest E-scooter ridership is reported at 4 a.m. (1482 E-scooter ridership) and at its highest at 12 p.m. (94,552 E-scooter ridership). The ridership at 12 p.m. is nearly 64 times greater than 4 a.m. The highest E-scooter incident was reported at 5 p.m. (14 E-scooter incidents), and lowest at 4 a.m. and 6 a.m. (1 E-scooter incident). Traffic incidents occurred the most at 9 p.m. (1329 traffic incidents) and lowest at 9 a.m. (113 traffic incidents). Traffic incidents at 9 p.m. are nearly 12 times higher than 12 h before.

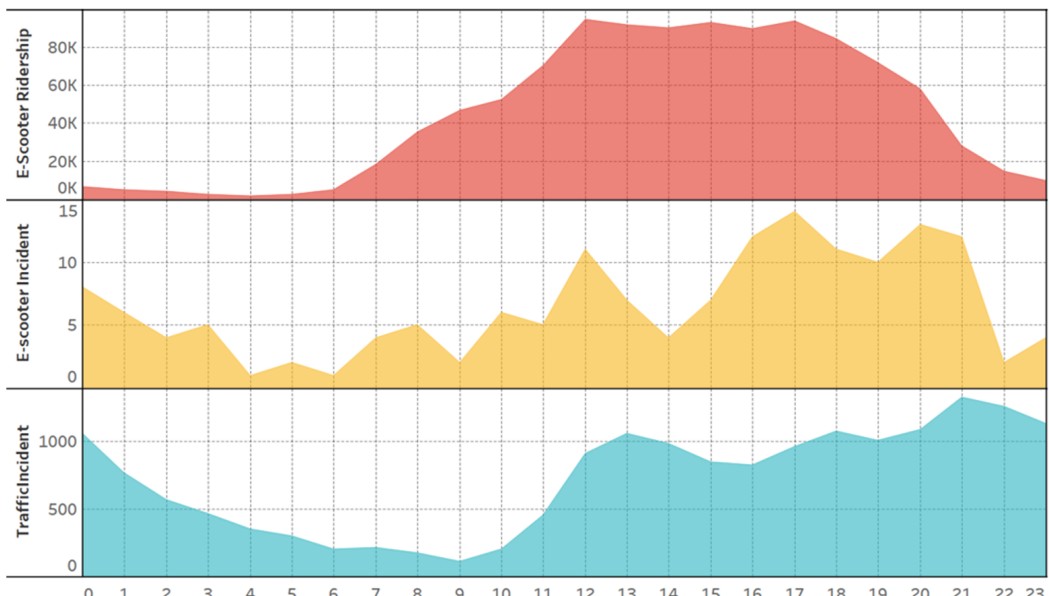

**Figure 3.** Hourly pattern of E-scooter ridership and incidents in Austin, TX, between 1 September 2018 and 30 November 2018.

Figure 4 describes the correlation matrix of hourly grouped E-scooter ridership, incidents, and traffic incidents. E-scooter incident shows a moderate positive correlation with E-scooter ridership and traffic incidents of correlation value of 0.61 and 0.59, respectively. Traffic incidents showed a weak positive correlation with E-scooter ridership with a correlation value of 0.33. The slope of traffic incidents and E-scooter ridership is lower than the slope calculated by E-scooter incidents with its ridership and E-scooter incidents with traffic incidents.

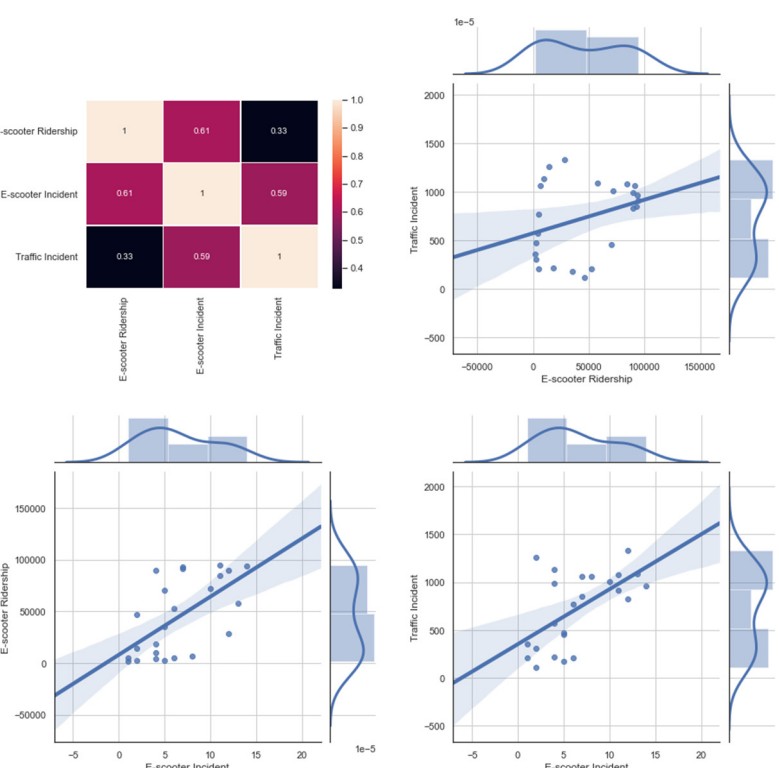

**Figure 4.** Correlation matrix of E-scooter ridership, E-scooter incident, and traffic incident.

*4.2. Local Moran's I Result*

Figure 5 describes the result of the local Moran's I analysis. Local Moran's I determines where spatial clusters and outliers are located, whether high or low in values. Spatial clusters include high-high clusters with high-value neighborhoods and low-low clusters with low-value neighborhoods. Similarly, spatial outliers include high-low and low-high. The former refers to a low-value neighborhood with a high-value outlier. The latter type indicates high-value neighborhoods with a low-value outlier [29].

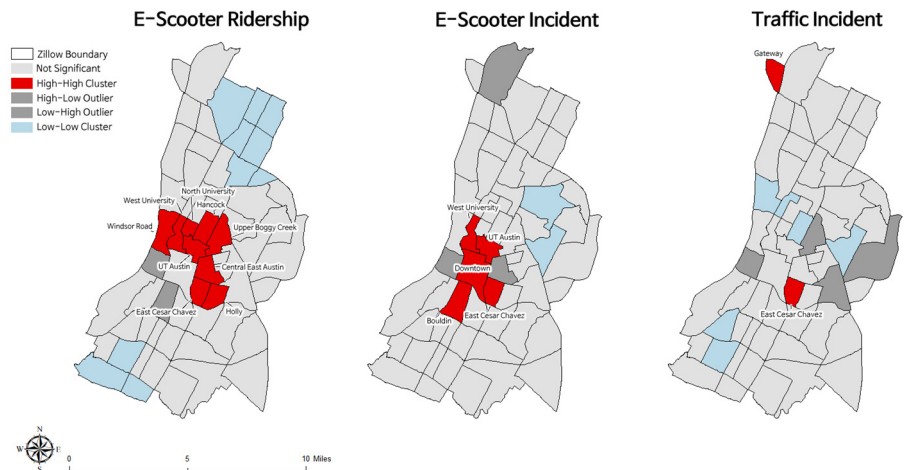

**Figure 5.** Local Moran's I result of E-scooter ridership, E-scooter incident, and traffic incident in Austin, TX, between 1 September 2018 and 30 November 2018.

The result shows that spatial correlation exists between E-scooter ridership, E-scooter incidents, and traffic incidents by forming high-high and low-low clusters with statistical significance. Specifically, a high degree of clustering happens near the university neighborhood (West University, North University, UT Austin, Hancock, Upper Boggy Creek)

and downtown areas (Central East Austin, East Cesar Chavez, Holly). Both northern and southern neighborhoods show a lower degree of clusters. E-Scooter incidents show a higher degree of clusters near downtown (Downtown, East Cesar Chavez, Bouldin) and the university neighborhood (West University, UT Austin), and a lower degree of the cluster was shown in the northern-central neighborhood. Traffic incidents show a high degree of the cluster in the downtown neighborhood (East Cesar Chavez) and part of the northern neighborhood (Gateway). East Cesar Chavez, a downtown neighborhood, has high degrees of E-scooter ridership, E-scooter incidents, and traffic incidents, which explains the spatial correlation between variables.

### 4.3. Independent Z-Test Result

Previous spatial statistics analysis shows spatial similarity and dissimilarity between E-scooter incidents and traffic incidents in different neighborhoods. We conducted Z-tests to identify how land use, built environment, and street characteristics correlate with these incidents. A total of 1566 street networks were used for the Z-tests. Results show that 149 street networks (10%) were associated with E-scooter incidents and 1417 street networks (90%) related to traffic incidents. The Z-test result is described in Table 3. The result shows a statically significant difference between the physical environment of E-scooter incidents and traffic accidents. Adjacent residential land use ($p < 0.01$), length of the street ($p < 0.001$), primary street type ($p < 0.05$), residential street type ($p < 0.01$), the total number of street nodes ($p < 0.001$), and a total number of traffic signals ($p < 0.001$) were statistically significantly different.

**Table 3.** Z-test result.

| Variable | | Z | p |
|---|---|---|---|
| Land Use and Built Environment | Residential | −2.5842 | 0.0098 ** |
| | Commercial | 0.5699 | 0.5687 |
| | Office | 0.8311 | 0.4059 |
| | Industrial | −0.1635 | 0.8701 |
| | Public | 0.6657 | 0.5056 |
| | OpenSpace | 0.1480 | 0.8824 |
| | Transportation Related | 0.5043 | 0.6140 |
| Street Edges | Length | $-1.60 \times 10^2$ | 0.0000 *** |
| | One-way | 1.5768 | 0.1148 |
| | Link | 0.0701 | 0.9441 |
| | Living Street | 0.0779 | 0.9379 |
| | Motorway | 0.0329 | 0.9737 |
| | Primary | 2.1168 | 0.0343 * |
| | Residential | −2.9462 | 0.0032 ** |
| | Secondary | 0.1256 | 0.9001 |
| | Territory | 0.0000 | 1.0000 |
| | Trunk | −0.041 | 0.9673 |
| | Unclassified | 0.4718 | 0.6371 |
| Street Nodes | Total Nodes | 17.4173 | 0.0000 *** |
| | Motorway Junction | 0.0906 | 0.9278 |
| | Traffic Signal | 4.7794 | 0.0000 *** |
| | Turning Circle | −0.3029 | 0.7620 |
| | Stop Sign | 0.3117 | 0.7553 |

(* $p < 0.05$, ** $p < 0.01$, *** $p < 0.001$).

## 5. Discussion

Micro-mobility programs such as E-scooter sharing are becoming increasingly important and relevant as alternative transportation modes for short distance travels. As more users are willing to use micro-mobility during the COVID-19 pandemic and beyond, it is critical to identify and analyze micro-mobility injuries and crashes to secure safe urban environments. Understanding E-scooter incidents in a street network perspective helps create a better understanding of the characteristics of E-scooter incidents and their occurrences.

Our study conducted three different phases. First, the time series analysis of E-scooter ridership, E-scooter incidents, and traffic incidents was conducted to identify temporal patterns throughout time. Second, local Moran's I was used to explore the spatial correlation between three variables and explain the spatial pattern. Finally, the present study conducted an independent Z-test to identify the statistical difference between E-scooter incidents and traffic incidents from a street network perspective.

Key findings include the following. First, E-scooter ridership and traffic incidents share similar patterns throughout time; however, E-scooter incidents hardly shared similar patterns, but instead fluctuated. Nevertheless, E-scooter incidents do seem to occur more when there is more significant E-scooter ridership. Second, local Moran's I verified past research findings that E-scooter usage tends to occur more near downtown and university neighborhoods [9]. These neighborhoods were also found to have a relatively greater degree of E-scooter incidents than others. Moreover, the same downtown neighborhood was spatially correlated with the first two variables, which explains the likelihood of more E-scooter ridership and more E-scooter and traffic incidents. Third, independent Z-test results showed a statistical difference between E-scooter incidents associated with the street network, with traffic incidents concerning one. Specifically, residential land use, length of the street, primary and residential street type, number of street nodes, and traffic signals were statistically significant. These results demonstrate that characteristics of E-scooter incidents do not fully correlate with traffic incidents, and surrounding built environments concerning crashes differs. Therefore, these results should be carefully considered for devising planning initiatives to create safe urban environments.

## 6. Conclusions

The present study contributes to shared micro-mobility research in several ways. First, the study identified and verified previous findings that explain E-scooter ridership. Second, we used an unexplored dataset and suggested a new research framework for understanding E-scooter incidents or crashes. Third, this study is one of the first E-scooter safety studies that identified the difference between E-scooter incidents and general traffic incidents from the street network perspective. This study can provide planners with the characteristics of E-scooter incidents and improve understanding of contexts of injuries and crashes.

As there are statically significant built environment differences between E-scooter and traffic incidents in general, planners should avoid implementing the same traditional measures to mitigate E-scooter incidents. Moreover, a measure to control other micro-mobility incidents, such as bike-sharing, cannot be the ultimate panacea since E-scooter crashes are not completely interchangeable [7]. Instead, E-scooter incidents should be individually explored and analyzed. E-scooter injuries seem to commonly occur on sidewalks, neighborhood streets, and major roadways, and the time of use and travel speed affects the severity of E-scooter injury [15,18]. E-scooter incidents tend to occur near streets adjacent to traffic signals and in primary street types compared to traffic incidents. In this case, a system that alerts E-scooter users to pay closer attention near sidewalks and increased traffic signal visibility may help reduce E-scooter incidents. Clear and visible symbols or colors on roads may help resolve safety issues. As smart crosswalk and signal systems are being implemented in cities, a system that identifies and control E-scooter users automatically and controls the traffic signal duration may prevent potential E-scooter

incidents. Exploring the application of new innovative technology remains a task for future smart cities.

For future study, we suggest exploring the factors associated with E-scooters by surveying E-scooter users. For instance, measuring the effects of wearing protective gear on the severity of the injury is critical. Because the rate of using helmets is mainly found to be low for shared micro-mobility users [13,30]. Alternatively, as shared micro-mobility is more visible with other micro-mobilities in streets, evaluating the present risk factors will be helpful to apprehend shared micro-mobility safety [30]. However, the lack of traffic and crash data related to E-scooters is the main obstacle for future E-scooter safety studies. Perhaps applying crowdsourcing tools to obtain public opinions may help gather local safety data and bring valuable insight to planning practices [3].

The present study has several limitations. First, there are no general public E-scooter incidents data. Therefore, the research scope of the present study is limited. Second, land use and built environment factor gathered using round buffer, so the result may differ depending on its size. Third, land use data are based on 2012 records and may be outdated. Fourth, the limitation of using Zillow neighborhood boundaries is that the size is hardly big enough to cover up the entire City of Austin. Though the boundary falls into shared micro-mobility service areas, the geographic scope is limited. Future studies should consider these limitations.

**Author Contributions:** Conceptualization, J.J., S.B., and S.J.C.; methodology, J.J., S.B., and S.J.C.; validation, J.J., S.B., and S.J.C.; formal analysis, S.B. and S.J.C.; resources, S.B. and S.J.C.; data curation, S.J.C.; writing—original draft preparation, S.B. and S.J.C.; writing—review and editing, J.J., S.B., and S.J.C.; visualization, S.J.C.; supervision, J.J.; project administration, J.J.; funding acquisition, J.J. All authors have read and agreed to the published version of the manuscript.

**Funding:** This work has been funded through the UT Good System Grand Challenge and the USDOT Cooperative Mobility for Competitive Megaregions University Transportation Center at The University of Texas at Austin.

**Institutional Review Board Statement:** Not applicable.

**Informed Consent Statement:** Not applicable.

**Data Availability Statement:** The data described in this article are openly available at City of Austin Open Data Portal, https://data.austintexas.gov/Transportation-and-Mobility/Shared-Micromobility-Vehicle-Trips/7d8e-dm7r (accessed on 7 April 2021); https://data.austintexas.gov/Transportation-and-Mobility/Real-Time-Traffic-Incident-Reports/dx9v-zd7x/data (accessed on 7 April 2021), and Patch, https://patch.com/texas/downtownaustin/scooter-crash-analysis-gives-glimpse-injuries-toll-austin (accessed on 7 April 2021).

**Acknowledgments:** The authors declare that they have no known competing financial interests or personal relationships that could have appeared to influence the work reported in this paper.

**Conflicts of Interest:** The authors declare no conflict of interest.

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
