# Peer review of "Understanding E-Scooter Incidents Patterns in Street Network Perspective: A Case Study of Travis County, Texas"

_sustainability, doi:10.3390/su131910583_

Round 1

Reviewer 1 Report

This paper reviews the safety of e-scooters in Travis Co, TX. A few questions/comments for the authors: 

Methodology - What are the limitations of the authors methodology, including limitations of using Zillow neighborhoods? 

Conclusion - Do the authors have any recommendations for additional research? 

Reference #4 should be corrected. The authors are Shaheen and Cohen. 

Some additional resources that may be helpful include: 

Micromobility Safety - https://transweb.sjsu.edu/research/Bikesharing-and-Bicycle-Safety

Reviewer 2 Report

This article presents an spatial analysis of dockless electric scootersharing ridership and traffic incidents in comparison to car traffic accidents in the city of Austin, Texas. The article is lacking thorough documentation of the methods applied and explanation of their value for producing novel insights regarding scootersharing safety and related policy.

The article does not fully detail the data nor the methods used in the analysis. Basic details about how the Patch e-scooter data was collected are not provided, including details about the sampling method which is necessary for the reader to understand any potential biases of the scooter incident data. There is also no discussion of the source or nature of the scooter ridership data nor the land use and built environment data. While Table 1 lists the variables  and descriptive statistics for several variables, the units of analysis are not clear. It is unclear, for example, what the ratios for the land use and built environment variables consist of, why the street edge variables are dummies while the street node variables are counts.

A ‘time series analysis’ is presented, which consists of three graphs of hourly data during a three-month time period for which the method of aggregation is not described. The graphs are described qualitatively without further analysis. The brief discussion of the time series that is provided seems to compare scootersharing ridership with traffic incidents without acknowledging the similarity of the two categories of incidents. More rigorous analysis is recommended to provide meaningful insights from the time series data.

A Local Moran’s analysis is presented without explanation of this methodology. The reader thus has no context with which to understand the results of this analysis nor its appropriateness for the research questions pursued.

Lastly, the methodology used for the statistical tests is not clearly explained. It is difficult to understand the meaning of the variables used in the Z-test as well as the justification for the method applied. The variables are described as averages of incidents associated street networks – does this refer to street nodes or edges? How are the other data applied to these variables?
